# Progressive Autoregressive Video Diffusion Models

## Abstract

Current frontier video diffusion models have demonstrated remarkable results at generating high-quality videos. However, they can only generate short video clips, normally around 5 seconds or 120 frames, due to computation limitations during training. In this work, we show that existing models can be naturally adapted to autoregressive video diffusion models without changing the architectures. Our key idea is to assign the latent frames with progressively increasing noise levels rather than a single noise level. Thus, each latent can condition on all the less noisy latents before it and provide condition for all the more noisy latents after it. Such progressive video denoising allows our models to autoregressively generate frames without quality degradation. We present state-of-the-art results on long video generation at 1 minute (1440 frames at 24 FPS). Our results are available at this anonymous url: `https://progressive-autoregressive-vdm.github.io/`.

## 1 Introduction

Video diffusion models have recently demonstrated remarkable success in generating high-quality video content across a variety of applications. These models are capable of synthesizing realistic video sequences that are increasingly indistinguishable from real-world footage. However, despite their impressive results, current video diffusion models are constrained by a significant limitation: they can only generate videos of relatively short duration, typically up to about 10 seconds. This temporal restriction leads to challenges for broader applications that require longer, more continuous video outputs, highlighting the need for further research and innovation to extend the capabilities of these models.

One straightforward way to generate longer video is averaging the predicted noise at each time step across latent segments Hu (2024); Tian et al. (2024a). However, such methods are still limited by the memory, and fail to preserve long-term consistency. On the other hand, several approaches Ho et al. (2022b); Henschel et al. (2024); Blattmann et al. (2023); Brooks et al. (2024); Gao et al. (2024) have been proposed to address the challenge of generating longer videos with diffusion models, which iteratively generates video clips, with each subsequent clip conditioned on the final frames of the previous one. One variant Ho et al. (2022b) directly puts the conditioning frames into the input frames, replacing the noisy frames, while another variant Brooks et al. (2024); Gao et al. (2024) additionally adds noise to the conditioning frames. Both methods have proven effective at producing smooth pixel-level transitions between clips, e.g., no temporal jittering between original video and extended video. However, these approaches struggle to accurately preserve secondary motion attributes, such as motion velocity and acceleration, leading to unnatural or inconsistent movement in longer sequences. Additionally, since these methods are still constrained by a maximum extension length, e.g, around 10 seconds, they must be applied repeatedly in a windowed fashion for generating substantially longer videos. This repetitive application amplifies the aforementioned issues, potentially increasing inaccuracies in motion dynamics and transitions, and accumulating errors that causes the video to eventually diverge across the entire video.

We propose an autoregressive video diffusion model that denoises video frames in a *progressive* manner, allowing for both high-quality video content extension and smooth motion generation. The core innovation of our method lies in the denoising process: instead of applying a single noise level across all frames used in traditional video generation or extension diffusion models, we progressively

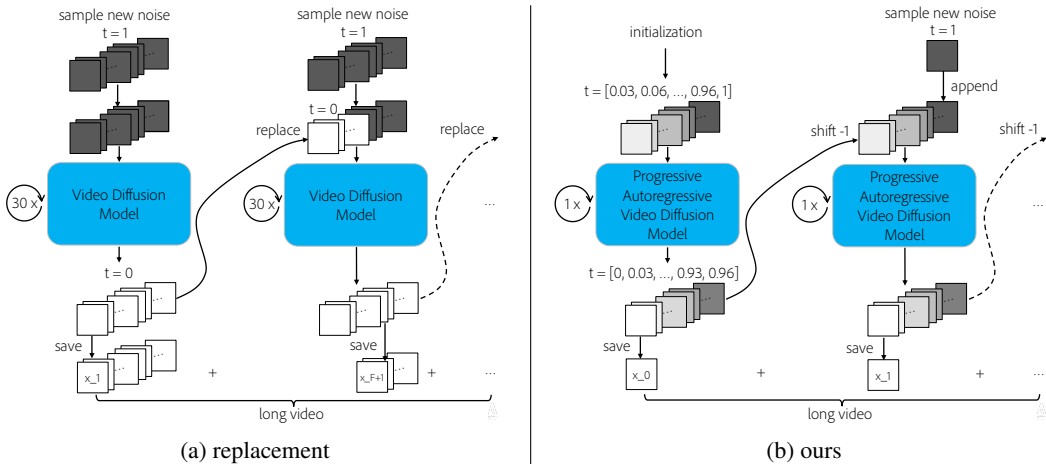

Figure 1: Comparison of autoregressively applying video diffusion models with replacement methods (left) vs. our progressive autoregressive video diffusion models (right).

increase the noise levels across the frames during denoising. This approach improves the temporal transitions in extended frames, resulting in more natural motion and better consistency. Our method can be easily implemented by changing the noise scheduling of pre-trained video diffusion models, either UNet- or DiT-based backbone, without changing the original model architecture. Our inference procedure can work training-free, if the model has gone through masked pre-training (Zheng et al., 2024), which allows the model to disentangle the noise levels from the latent frames and learning a per-frame noise level embedding. If not, we can simply finetune the model to adapt to the new progressive noise level distribution. Our method, either training-free or with finetuning, enables generating of videos up to one minute in length (1440 frames) without noticeable degradation in quality. Moreover, the additional computational cost at inference time is minimal comparing to previous work Wang et al. (2023) having overlapped regions to generate, making this approach efficient for practical use in long video generation.

To facilitate future research, we will release our training and inference code based on Open-Sora (Zheng et al., 2024). We will also release the model weights after we train our Open-Sora-based model on open datasets.

## 2 BACKGROUND

### 2.1 VIDEO DIFFUSION MODELS

Diffusion models (Ho et al., 2020) are generative models that learn to generate samples from a data distribution $\mathbf{x} \sim X$ through an iterative denoising process. During training, data samples are first corrupted using the forward diffusion process $q(\mathbf{x}^t|\mathbf{x})$, which adds Gaussian noises of level $t \in [0, 1]$ to the sample.

$$q\left(\mathbf{x}^{t_i}\big|\mathbf{x}^{t_{i-1}}\right) = \mathcal{N}(\mathbf{x}^{t_i}; \sqrt{1 - \beta^{t_i}}\mathbf{x}^{t_{i-1}}, \beta^{t_i}I), \quad q\left(\mathbf{x}^{t_i}\big|\mathbf{x}^{t_0}\right) = \mathcal{N}(\mathbf{x}^{t_i}; \sqrt{\bar{\alpha}^{t_i}}\mathbf{x}^{t_0}, (1 - \bar{\alpha}^{t_i})I) \quad (1)$$

The diffusion model, with parameters $\theta$, can be trained with a mean squared error loss (Ho et al., 2020).

At sampling time, given the number of sampling steps $S$, we have a sampling noise schedule $\boldsymbol{t} = \{t_0, t_1, ..., t_{S-1}\}$, where $0 = t_0 \leq t_1 \ldots \leq t_{S-2} \leq t_{S-1} = 1$. Starting from $\mathbf{x}^1 \sim \mathcal{N}(0, \boldsymbol{I})$, we iterative apply the denoising process; given the data with the current noise level $t$, we can obtain the data with the previous noise level $t - 1$ from the following conditional distribution

$$p_\theta\left(\mathbf{x}^{t_{i-1}}\big|\mathbf{x}^{t_i}\right) \quad (2)$$

Among the samples $\mathbf{x}^1, \mathbf{x}^{t_{S-2}}, \ldots, \mathbf{x}^{t_1}, \mathbf{x}^0$, the last sample $\mathbf{x}^0$ is the clean data.

Video diffusion models (Ho et al., 2022b) are diffusion models that consider video data $\mathbf{x}_{0:F-1} = \{\mathbf{x}_0, \mathbf{x}_1, \ldots, \mathbf{x}_{F-1}\}$ that consists of $F$ image frames $\mathbf{x}_i$. The same forward diffusion process and the

denoising process can be applied by treating all the frames as one entity, ignoring the correlation among the frames .

## 2.2 Long Video Generation via Replacement

Video diffusion models can only generate short video clips, because they are only trained on videos with a limited length $F$ due to GPU memory limit. When adapted to generate $L > F$ frames zero-shot at inference time, their generation quality substantially degrades (Qiu et al., 2024).

One simple solution is to autoregressively apply video diffusion models, generating each video clip while conditioning on the previous clip. Specifically, given $E < F$ clean frames $\mathbf{x}_{0:E}^0$ as condition, Ho et al. (2022b); Song et al. (2020b) proposed *the replacement method* to sample from the conditional distribution

$$p_\theta \left( \mathbf{x}_{0:E}^{t_{i-1}}, \mathbf{x}_{E:F-1}^{t_{i-1}} \middle| \mathbf{x}_{0:E}^{t_i}, \mathbf{x}_{E:F-1}^{t_i} \right) \tag{3}$$

where $\mathbf{x}_{0:E}^{t_i}$, the exact conditioning frames noised via the forward process, directly replaces the sampled frames at each denoising step. We will refer to this method as the *replacement-with-noise* method

On the other hand, Zheng et al. (2024); Gao et al. (2024); Blattmann et al. (2023) conditions clean frames directly at the beginning of the current video clip without adding noise as

$$p_\theta \left( \mathbf{x}_{0:E}^0, \mathbf{x}_{E:F}^{t_{i-1}} | \mathbf{x}_{0:E}^0, \mathbf{x}_{E:F}^{t_i} \right) \tag{4}$$

We will refer to this method as the *replacement-without-noise* method. Both the *replacement-with-noise* method and the *replacement-without-noise* method allows a video diffusion model to autoregressively generate video frames by conditioning on previous frames. We consider them as baselines in our experiments in Sec. 4.2.

## 3 Progressive Autoregressive Video Diffusion Models

Although existing video diffusion models (Zheng et al., 2024) can only generate videos up to a limited length (e.g. 5 seconds or 120 frames), we show that they can be naturally adapted to become autoregressive video diffusion models without changing the architectures. We achieve this by proposing a per-frame noise schedule, which is inspired by (Chen et al., 2024). During training, we finetune pre-trained video diffusion models to adapt to such noise schedule; during sampling, our models adopt such noise schedule and can thus autoregressively generate video frames.

### 3.1 Progressive Video Denoising

Inspired by (Chen et al., 2024), we assign progressively increasing noise levels to video frames being denoised. Autoregressive video diffusion models sample from the following conditional distribution

$$p(\mathbf{x}_0^{t_{i-1}}, \mathbf{x}_1^{t_i}, ..., \mathbf{x}_{F-2}^{t_{i+F-3}}, \mathbf{x}_{F-1}^{t_{i+F-2}} | \mathbf{x}_0^{t_i}, \mathbf{x}_1^{t_{i+1}}, ..., \mathbf{x}_{F-2}^{t_{i+F-2}}, \mathbf{x}_{F-1}^{t_{i+F-1}}) \tag{5}$$

where the frames $\mathbf{x}_f, f \in [0, F)$ have progressively increasing noise levels $t_{i-1} < t_i < t_{i+1} < ..., t \in [0, T)$.

By assigning individual noise levels $t_f$ to each frame $\mathbf{x}_f$, we are effectively using a single set of model parameters $\theta$ to jointly model diffusion process of each frame $\mathbf{x}_f$, which has a scalar noise level $t_f$ like regular diffusion models. Thus, the foundations of diffusion models, including training and sampling, can still apply to our progressive video diffusion models. Figure 2 provides an illustration comparing the proposed noise level approach with the previous replacement method.

Our progressive video denoising process gradually establishes correlation among consecutive frames. Given some existing video frames as conditioning, it is challenging for video diffusion models to produce temporally consistent extensions frames from newly sampled noisy frames (Qiu et al., 2024). In contrast to the *replacement* methods where numbers of noisy frames are inferred together, our progressive noise facilitates modeling a smoother and more consistent temporal transition,

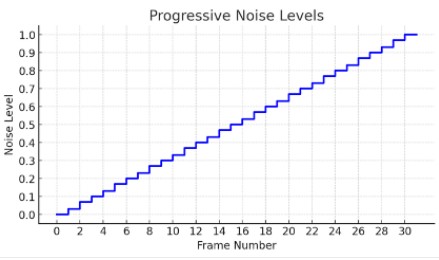 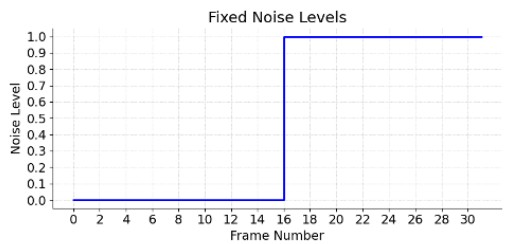

Figure 2: Comparison of noise levels of ours vs. the replacement without noise method.

encouraging the later frames with higher uncertainty to follow the patterns of the earlier and more certain frames.

Another perspective is to consider a toy example of learning to fit a single long video of $L$ frames, using video diffusion models with a limited window length $F$. The model needs to fit any subset of $F$ frames from $L$ total frames at training time, and being able to generate cohesive $L$ frames at inference time. The neighboring data points in the training set of our method, i.e. eq. (6) are exactly one inference step apart from each other. Such formulation establishes a consistent denoising process during training and inference, whereas the models in replacement methods are trained for a single denoising step in every iteration, making it harder to fit.

### 3.2 Autoregressive Generation

For simplicity, we consider the following instantiation of eq. (5) where $F = S$

$$p(\mathrm{x}_0^0, \mathrm{x}_1^{T/S}, ..., \mathrm{x}_{F-2}^{(S-2)T/S}, \mathrm{x}_{F-1}^{(S-1)T/S} | \mathrm{x}_0^{T/S}, \mathrm{x}_1^{2T/S}, ..., \mathrm{x}_{F-2}^{(S-1)T/S}, \mathrm{x}_{F-1}^{T}) \tag{6}$$

We notice that after one sampling step, we obtain a clean frame $\mathrm{x}_0^0$. By removing the clean frame and appending a new noisy frame $\mathrm{x}_{F-1}^T$ at the end, our frames have the same input noise levels $t = T/S, 2T/S, ..., (S-1)T/S, T$ again. Alg. 1 describes the inference procedure of our progressive autoregressive video diffusion models.

---

**Algorithm 1** Inference procedure of autoregressive video diffusion models

---

**Require:** Initial video sequence $\mathbf{x} = \{\mathrm{x}_0, \mathrm{x}_1, ..., \mathrm{x}_{F-1}\}$, total noise level $T$, number of inference steps $S$, and number of frames $F$

$\quad \boldsymbol{t}_{0:F}$ $\qquad\qquad\qquad\qquad\qquad\qquad$ ▷ Initialize progressively increasing noise levels

$\quad \epsilon \sim \mathcal{N}(0, \boldsymbol{I})$

$\quad \mathbf{x}^{\boldsymbol{t}} = add\_noise(\mathbf{x}, \boldsymbol{t}, \epsilon)$

$\quad$ **for** each autoregressive generation step $i = 1, 2, \ldots, N$ **do**

$\quad\quad \left( \mathrm{x}_0^{t_0}, \mathrm{x}_1^{t_1}, \ldots, \mathrm{x}_{F-1}^{t_{F-1}} \right) \sim p \left( \mathrm{x}_0^{t_0}, \mathrm{x}_1^{t_1}, \ldots, \mathrm{x}_{F-1}^{t_{F-1}} | \mathrm{x}_0^{t_1}, \mathrm{x}_1^{t_2}, \ldots, \mathrm{x}_{F-1}^{t_F} \right)$

$\quad\quad \mathbf{x} = \left\{ \mathrm{x}_0^{t_1}, \mathrm{x}_1^{t_2}, \ldots, \mathrm{x}_{F-1}^{t_F} \right\}$ $\qquad$ ▷ Remove the clean frame and append a new noisy frame

$\quad$ **end for**

---

**Variable Length** While the above design allows for autoregressively extending a video of length $F$, we can easily accommodate it for text-to-video generation without any given starting frames. In addition, the noisy frames remaining in the frame sequence are discarded after the end of the autoregressive inference, which can cause wasted computing resources and inaccurate handling of the ending of text prompt. To address the above, we propose to extend the base design in eq. (6) and Alg. 1 to add an initialization stage and an termination stage. During initialization, we start with one frame, denoise it for one step, append a noisy frame with $t = T$ without saving and removing any frames, and finally reach $F$ frames with the progressive noise levels described in eq. (6). During termination, we start with $F$ frames with progressive noise levels in eq. (6), denoise them for one step, save and remove the first frame with $t = 0$ without appending new noisy frames, and finally reach 1 frames with $t = 0$. We train the model accordingly on variable input video length and the corresponding noise levels.

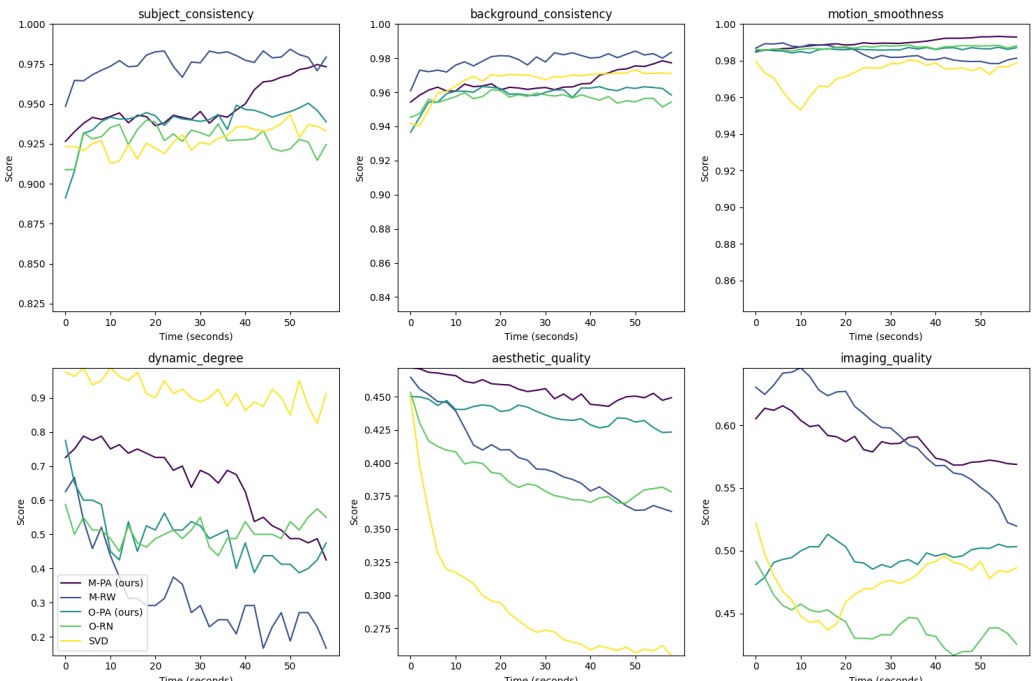

Figure 3: VBench (Huang et al., 2024) scores of generated videos over the 60-second duration, averaged over 80 videos from our testing set. The scores are computed on 30 2-second clips. Our models *M*-PA and *O*-PA can best maintain the level of dynamic degree, aesthetic quality, and imaging quality over time compared to other baselines. Notably, baselines that use the same model as ours, *M*-RW and *O*-RN, both exhibit substantial drop in dynamic degree, aesthetic quality, and imaging quality.

## 3.3 Training

We finetune the base video diffusion model by modifying the diffusion timesteps during training. Regular diffusion model training involves sampling a timestep $t \in [0, T)$ and adding noise with level $t$. To achieve progressive noise level in eq. (5), the noise level is changed from a single scalar $t$ to a vector $\boldsymbol{t} = \{t_0, t_1, \ldots, t_{F-1}\}$ corresponding to each frame. In our experiment, we observed that using a simple linear noise schedule yielded satisfactory results for all reported experiments. During training, the noise level of $\boldsymbol{t}$ is perturbated by a random shift $\delta$ to preserve the coverage of the full diffusion timestep range $[0, T)$ (Song et al., 2020a). $\delta = 0.4\epsilon(t_i - t_{i+1}), \epsilon \sim \mathcal{N}(0, \boldsymbol{I})$ is randomly sampled for each training iteration and remains constant for all $t_i$ within that iteration.

## 4 Experiments

### 4.1 Implementation

**Base model** We implement autoregressive video diffusion models by fine-tuning from pre-trained models. Specifically, we use two latent video diffusion models based on the diffusion transformer architecture (Peebles & Xie, 2023; Brooks et al., 2024): Open-Sora (Zheng et al., 2024) and a modified variant of Open-Sora. We will denote them as *O* and *M* respectively. Both models are latent diffusion models, utilizing a corresponding 3D VAE that encodes 16 video frames into 5 latent representations. *O* generates outputs at 240x424 resolution at 24 FPS with 30 inference steps. *M* produces results at 176x320 resolution at 24 FPS with 50 inference steps. We also consider two baseline autoregressive video generation methods, *replacement-with-noise* (RW) and *replacement-without-noise* (RN), which are implemented on *M* and *O*.

We train $M$ on our progressive noise levels, as discussed in Sec. 3.3. We denote this model as $M$-PA (Progressive Autoregressive). We also train $M$ with the *replacement-with-noise* method (Sec. 2.2), which we will denote as $M$-RW. Starting from the same base model, $M$-RW is trained for 3 times for training steps compared to $M$-PA.

We implement our progressive video denoising sampling procedure (Sec. 3.2 and Alg. 1) on $O$, denoted as $O$-PA. We find that $O$-PA can directly adapt to our progressive noise levels training-free. We believe that this is because $O$ undergoes masked pre-training (Zheng et al., 2024), which allows it learn that the noise levels $t$ can be independent with respect to the latent frames.

**Training details** We train on captioned image and video datasets, containing 1 million videos and 2.3 billion image data. These data are licensed and have been filtered to remove not-safe-for-work content. We train on various of video length including $16, 32, ..., 176$ frames that correspond to $5, 10, ..., 55$ latents. The 55 latent frames length is derived by setting number of latent frames equal to 50 inference steps plus an additional chunk of latent frames, as discussed below. The shorter latent frame lengths $5, 10, ..., 50$ are used in the initialization and termination stages of our autoregressive generation process, as discussed in Sec. 3.2.

**Modification to the base model** To implement autoregressive video diffusion models on top of their base video diffusion models, we do not need to modify the base model architectures. Instead, we only need to modify the following in the model's forward, training, and inference procedures. In the inference and training procedures, we replace scalar diffusion timestep $t \in [0, T)$, from regular diffusion model training (Ho et al., 2022b; 2020), with a list of timesteps with length $F$, $\boldsymbol{t} = t_0, t_1, ..., t_{F-1}$. To accommodate this change, we also need to change how the model processes the diffusion timestep to get the timestep embedding. Our timesteps input has two dimensions, $B \times F$. We first merge the the two dimensions, pass it to the timestep embedding module, and reassemble the two dimensions, and finally broadcast the timestep embedding to the same size of the latents so they can be combined through addition, concatenation, or modulation (Peebles & Xie, 2023; Perez et al., 2018).

**Chunk** 3D VAE (Zheng et al., 2024) usually encode and decode video latent frame chunk-by-chunk. In our early experiments, we find that there is serious cumulative error when given each frame different noise levels and shift the window one frame at a time, causing the generated videos to diverge quickly after a few seconds. When looking at the videos closely, we notice that the cumulative error worsens after every chunk. This leads us to believe that the cumulative error is caused by not denoising a chunk of latent frames together. We resolve the problem by treating each chunk of latent frames as a single frame: they are assigned with the same noise level, and will be added and removed from the frame sequence together. Our ablation experiments on both $O$ and $M$ show that the chunked training and inference substantially improves the generation result. $O$ and $M$ both have latent chunks of 5 frames.

**Keeping clean frames available in temporal self-attention** The default design of the input and output frame sequences presented in Sec. 3.2 results in temporal jittering. This is because the clean frames that reaches $t = 0$ are immediately removed; as the later frames cannot attend to the previous clean frames, even though they are already at a low noise level, it is hard to achieve perfect temporal consistency with the previous clean frames. In practice, we always keep a chunk of clean $\mathbf{X}_{-1}^0$ latent frames in front of the noisy frames. This helps resolving frame-to-frame discontinuity.

### 4.2 LONG VIDEO GENERATION

**Baselines** We only compare to baselines that use the same model but different conditioning mechanisms from ours. For the two base models, $O$ and $M$, we consider two conditioning mechanisms that were used in (Zheng et al., 2024; Henschel et al., 2024; Gao et al., 2024; Ho et al., 2022b): replacing noise with conditioning frames or conditioning frames with noise.

**Benchmarks** We consider 6 metrics in VBench (Huang et al., 2024), subject consistency, background consistency, motion smoothness, dynamic degree, aesthetic quality, and imaging quality. Our testing set consists of 40 text prompts and the corresponding real videos, sampled from Sora (Zheng

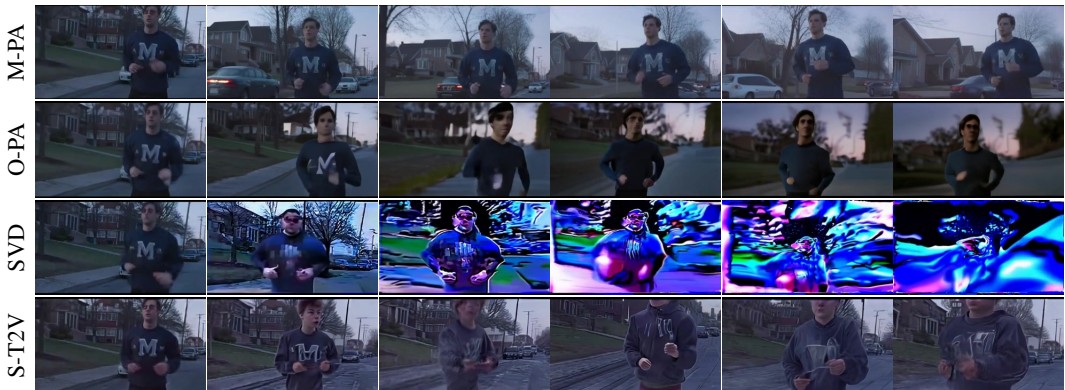

Figure 4: Qualitative comparison of ours M-PA, O-PA, SVD, StreamingT2V (S-T2V for short). Frames are evenly sampled from 1 minute long generated video.

et al., 2024) demo videos, MiraData (Ju et al., 2024), UCF-101 (Soomro, 2012), and LOVEU (Wu et al., 2023b;a). For each text prompt, we generate two 60-second videos, resulting in a total of 80 videos. We use these 80 videos from each model for both quantitative and qualitative results, unless specified otherwise. Due to computation resource limitations of sampling 1-minute long videos, we only obtained partial results from *M*-PA, including 48 videos from 24 text prompts.

**Quantitative Results**   Since our focus is on long video generation, we care about the video extension capability of the models rather than the text-to-short-video capability, we use the initial frames of the videos as the condition for all models, similar to the setting in (Henschel et al., 2024). *M*, *O* (Zheng et al., 2024), StreamingT2V (Henschel et al., 2024), and SVD (Blattmann et al., 2023) use 16, 17, 1, and 1 frames from the real video as the initial condition.

We present the average metrics for each model in Sec. 4.2. All models have obtained similar subject consistency, background consistency, and motion smoothness. Our *M*-PA obtains substantially better dynamic degree than the baseline *M*-RW. Our *M*-PA and *O*-PA also achieve better aesthetic quality and imaging quality than the baselines *M*-RW and *O*-RN.

In Fig. 3, we show the trend of scores over the 1-minute duration of videos for each model. All 6 models can maintain their subject consistency, background consistency, and motion smoothness scores over time. Our models *M*-PA and *O*-PA can best maintain the level of dynamic degree, aesthetic quality, and imaging quality over time compared to other baselines. Notably, baselines that use the same model as ours, *M*-RW and *O*-RN, both exhibit substantial drop in dynamic degree, aesthetic quality, and imaging quality.

Table 1: Quantitative comparison of two base models (*M* and *O*) with our progressive autoregressive video generation (PA) and two baseline methods *replacement-with-noise* (RW) and *replacement-without-noise* (RN), StreamingT2V (Henschel et al., 2024), and Stable Video Diffusion (SVD) (Blattmann et al., 2023).

| | Subject Consistency ↑ | Background Consistency ↑ | Motion Smoothness ↑ | Dynamic Degree ↑ | Aesthetic Quality ↑ | Imaging Quality ↑ |
|---|---|---|---|---|---|---|
| ***M*-PA** (ours) | 0.7923 | 0.8964 | 0.9896 | 0.8000 | 0.4726 | 0.5927 |
| *M*-RW | 0.8001 | 0.8851 | 0.9836 | 0.3958 | 0.4123 | 0.5961 |
| ***O*-PA** (ours) | 0.7656 | 0.8880 | 0.9859 | 0.5625 | 0.4582 | 0.5033 |
| *O*-RN | 0.7406 | 0.8820 | 0.9873 | 0.5750 | 0.4034 | 0.4464 |
| StreamingT2V | 0.8172 | 0.8916 | 0.9929 | 0.65 | 0.4264 | 0.5566 |
| SVD | 0.6102 | 0.8136 | 0.9724 | 0.9875 | 0.3019 | 0.4814 |

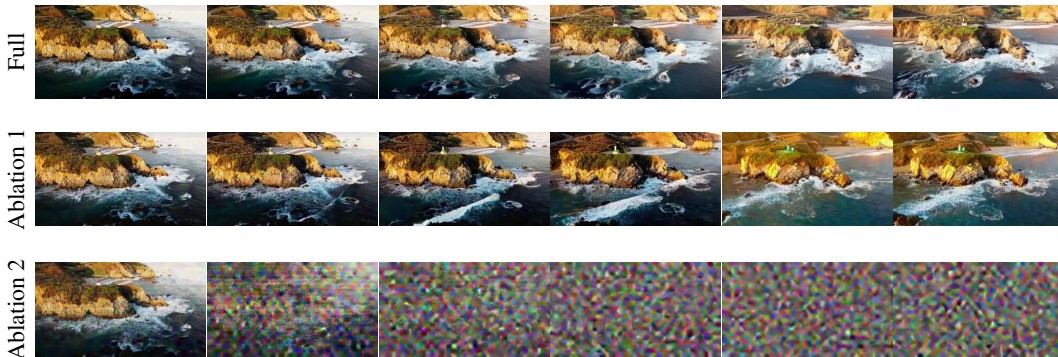

Figure 5: Qualitative comparison for ablation study. Full represents for our full solution based on M-PA, Ablation 1 is with chunk but without temporal self-attention. Ablation 2 is without both techniques. Frames are evenly sampled from a 16-second-long generated video.

**Qualitative Results**  We also show strength of our method with qualitative comparison results in Fig. 4. Both of our variants demonstrate strong performance in terms of frame fidelity and motion realism (e.g., running gestures in this example). M-PA outperforms O-PA due to additional fine-tuning with our proposed progressive noise levels, whereas O-PA simply inherits the pre-trained weights from Open-Sora (Zheng et al., 2024). In contrast, SVD shows severe artifacts that decreases frame validity, and StreamingT2V (S-T2V) suffers from cumulative errors, resulting in degraded video quality as the sequence length increases. For more qualitative results, please refer to our anonymous website, where we include all of the 80 videos from our testing set for all 6 models.

### 4.3 ABLATION STUDY

We conducted an ablation study on the M-PA model to evaluate the impact of Chunk (decoding video latents chunk-by-chunk) and Temporal Self-Attention (using an additional chunk of clean latents for temporal attention), as described in Section 4.1. Qualitative comparison has been shown in Figure 5. In Ablation 1, we observe that the absence of clean frames in the input sequence prevents noisy frames from attending to previous clean frames, resulting in poor performance over a long duration. This also causes frame-to-frame discontinuity, which is more noticeable in the supplementary anonymous webpage. In Ablation 2, not decoding the video chunk-by-chunk leads to severe cumulative errors, causing the video to diverge after only a few seconds.

### 5 RELATED WORKS

The field of long video generation has faced significant challenges due to the computational complexity and resource constraints associated with training models on longer videos. As a result, most existing text-to-video diffusion models Guo et al. (2023); Ho et al. (2022a;b); Blattmann et al. (2023) have been limited to generating fixed-size video clips, which leads to noticeable degradation in quality when attempting to generate longer videos. Recent works are proposed to address these challenges through innovative approaches that either extend existing models or introduce novel architectures and fusion methods.

Freenoise Qiu et al. (2024) utilizes sliding window temporal attention to ensure smooth transitions between video clips but falls short in maintaining global consistency across long video sequences. Gen-L-video Wang et al. (2023), on the other hand, decomposes long videos into multiple short segments, decodes them in parallel using short video generation models, and later applies an optimization step to align the overlapping regions for continuity. FreeLong Lu et al. (2024) introduces a sophisticated approach which balances the frequency distribution of long video features in different frequency during the denoising process. Vid-GPT (Gao et al., 2024) introduces GPT-style autoregressive causal generation for long videos.

More recently, Short-to-Long (S2L) approaches are proposed, where correlated short videos are firstly generated and then smoothly transit in-between to form coherent long videos. StreamingT2V Hen-

schel et al. (2024) adopts this strategy by introducing the conditional attention and appearance preservation modules to capture content information from previous frames, ensuring consistency with the starting frames. It further enhances the visual coherence by blending shared noisy frames in overlapping regions, similar to the approach used by SEINE Chen et al. (2023). NUWA-XL Yin et al. (2023) leverages a hierarchical diffusion model to generate long videos using a coarse-to-fine approach, progressing from sparse key frames to denser intermediate frames. However, it has only been evaluated on a cartoon video dataset rather than natural videos. VideoTetris Tian et al. (2024b) introduces decomposing prompts temporally and leveraging a spatio-temporal composing module for compositional video generation.

Another line of research focuses on controllable video generation Zhuang et al. (2024); Tian et al. (2024a); Hu (2024); Zhu et al. (2024) and has proposed solutions for long video generation using overlapped window frames. These approaches condition diffusion models using both frames from previous windows and signals from the current window. While these methods demonstrate promising results in maintaining consistent appearances and motions, they are limited to their specific application domains which relies heavily on strong conditional inputs.

## 6 DISCUSSION

In this work, we target long video generation, a fundamental challenge of current video diffusion models. We show that they can be naturally adapted to become progressive autoregressive video diffusion models without changing the architectures. With our progressive noise levels and the autoregressive video denoising process (Secs. 3.1 and 3.2), we obtain state-of-the-art results on long video generation at 1-minute long. Since our method does not require changing the model architectures, it can be seamlessly combined with many orthogonal works, paving the way for generating longer videos at higher quality, long-term dependency, and controllability.

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
