# OpenReview forum: "Progressive Autoregressive Video Diffusion Models"
_ICLR.cc/2025/Conference — ICLR 2025 Conference Withdrawn Submission_

### Official Review · Reviewer_xeo6 · 2024-11-01

**Soundness:** 2
**Presentation:** 2
**Contribution:** 1
**Rating:** 3
**Confidence:** 5

**Summary:**

This paper proposes to fine-tune existing video diffusion models (e.g., Open-Sora) for long video generation, where they employ a per-frame progressive noise scheduling. By doing so, the fine-tuned model lessens the error propagation problem and thus can generate long videos without frame-quality degradation. By fine-tuning the popular open-source video model called Open-Sora, they show the proposed fine-tuning scheme can generate up to 1-minute videos.

**Strengths:**

- The paper is generally well-written and easy to follow.
- Qualitatively, the result is better than other naive baselines.

**Weaknesses:**

- Per-frame noise scheduling has been recently introduced by many works, to name a few [1, 2, 3]. In particular, [1, 2] also deals with the training scheme of video diffusion models to generate long videos. In the current status, it seems there's no technical difference between these works and the proposed method. The only difference seems to be that this paper fine-tunes existing video diffusion models, which are not designed with per-frame scheduling---but for me, this is quite a marginal contribution to be accepted. If this is the main contribution of the paper, I think the paper should show the results with a variety of video diffusion models (not limited to open-sora) to validate that this fine-tuning scheme is not the special case of this model but rather can be applied to different diffusion model architecture, formulations (like noise scheduling --- flow-based, edm vs ddpm, etc), and so on.
- Lack of quantitative comparison. The paper fine-tunes the model using per-frame nose scheduling in [1], but it can be any other scheduling such as [1] or [3]. Specifically, even though the noise scheduling in [3] is for training-free long video inference, they can be used for fine-tuning before using them, similar to the authors' protocol to fine-tune the model on per-frame scheduling. How does the performance differ among three models: open-sora fine-tuned with [1], [2], and [3]?
- Dataset issue. It seems the authors curated the data for fine-tuning, but the paper doesn't include any details about this. Could the authors explain this more in detail?

[1] Rolling Diffusion Models, ICML 2024.
[2] Diffusion Forcing: Next-token Prediction Meets Full-Sequence Diffusion, NeurIPS 2024.
[3] FIFO-Diffusion: Generating Infinite Videos from Text without Training, NeurIPS 2024.

**Questions:**

- It seems all of the videos continue from videos generated by Sora. What about other videos (like videos generated with open-sora itself or the fine-tuned model)?
- Could the authors provide more details on dataset curation?

**Details Of Ethics Concerns:**

The dataset seems to be curated from the internet, but there are no details about it, even in the appendix. Specifically, there are only two sentences:

> We train on captioned image and video datasets, containing 1 million videos and 2.3 billion image data. These data are licensed and have been filtered to remove not-safe-for-work content.

---

### Official Review · Reviewer_nuv7 · 2024-11-03

**Soundness:** 2
**Presentation:** 2
**Contribution:** 2
**Rating:** 3
**Confidence:** 4

**Summary:**

This paper proposes an autoregressive video diffusion model by assigning different (progressive) noise schedule for different frames. The author shows that the method is capable of generating long and high-res videos while keeping temporal consistency.

**Strengths:**

The model shows promising results on generating long-term high resolution videos.

**Weaknesses:**

While this paper introduces a progressive noise schedule for window-based autoregressive video generation, there are several critical concerns that limit its contribution as a research paper.

1. **Lack of Novelty** (Existing previous work): The core idea of a progressive noise schedule across frames was previously proposed in the ICML 2024 paper [Rolling Diffusion Model](https://arxiv.org/pdf/2402.09470), which also employs this technique to enable autoregressive video generation. Although it’s possible that this prior work was unintentionally overlooked, it’s highly relevant and should be discussed in detail. Without addressing this, the contribution here appears more incremental than original. In my view, the primary contribution of this work lies in extending the experiments to high-resolution videos.

2. **Superficial Treatment of the Method**: Section 3 of the paper, which details the proposed method, spans only about one page and lacks a rigorous or in-depth analysis. While Sections 3.1 and 3.2 briefly explain the technique, Section 3.3 reads more like a subsection within the experimental section than an elaboration of the methodology. For a research paper, a deeper exploration of the theoretical insights and nuances of the approach would be expected.

3. **Misinterpretation of Related Work**: In Section 2, the authors discuss VDM and claim it uses a "replacement"-based autoregressive generation method. However, this is inaccurate—VDM’s primary focus is on improving temporal consistency in autoregressive generation using classifier-guidance (or reconstruction guidance), while “replacement” serves as an ablation study in the VDM paper. This misunderstanding should be corrected for clarity and accuracy when discussing related literature.

4. **Lack of Comprehensive Evaluation**: While the authors have adopted the newly proposed VBench as the primary evaluation benchmark, they do not provide quantitative results on mainstream metrics such as FVD, which is widely recognized and robust for evaluating temporal video prediction. Including FVD would enhance the paper’s credibility and make it more comparable to other video generation methods.

In sum, while the work offers a technically sound implementation, the limited novelty, insufficient methodological depth, and lack of comprehensive evaluation on standard metrics detract from its overall impact as a research paper.

**Questions:**

See weakness above.

---

### Official Review · Reviewer_KGVn · 2024-11-03

**Soundness:** 3
**Presentation:** 3
**Contribution:** 3
**Rating:** 5
**Confidence:** 4

**Summary:**

This paper shows that existing models can be naturally adapted to autoregressive video diffusion models without changing the architectures.  The authors assign the latent frames with progressively increasing noise levels rather than a single noise level. And each latent can condition on all the less noisy latents before it. The proposed method avoids quality degradation.

**Strengths:**

1. This paper proposes an autoregressive video diffusion model that denoises video frames in a progressive manner, allowing for both high-quality video content extension and smooth motion generation.
2.  The method can be easily implemented by changing the noise scheduling of pre-trained video diffusion models.
3. The additional computational cost at inference time is not large.

**Weaknesses:**

1. Some baseline models should be added. For example, VideoCrafter2, T2V-Turbo, Open-Sora.
2. In figure 2, has the authors tried different noise increasing schedule? And what's the influence on model performance?
3. Is there any model complexity analysis? For example, #params, FLOPs.

**Questions:**

See weakness.

---

### Official Review · Reviewer_2p7H · 2024-11-04

**Soundness:** 2
**Presentation:** 3
**Contribution:** 2
**Rating:** 5
**Confidence:** 4

**Summary:**

This paper proposes a novel approach called Progressive Autoregressive Video Diffusion Models (PA-VDM) to address the limitations of existing video diffusion models in generating long videos. Instead of using a single noise level for all frames, PA-VDM assigns increasingly higher noise levels to frames during the denoising process. This allows each frame to condition on all previous frames with lower noise levels and provide information for future frames with higher noise levels.

**Strengths:**

1. PA-VDM extends the capabilities of existing video diffusion models to generate longer videos, up to 1 minute in length (1440 frames at 24 FPS), without compromising quality. This is achieved by assigning latent frames with progressively increasing noise levels, allowing for autoregressive generation without degradation.
2. PA-VDM maintains temporal consistency throughout the generated video, ensuring smooth transitions and realistic motion dynamics. This is in contrast to other methods that struggle to preserve secondary motion attributes like velocity and acceleration, leading to unnatural movement.
3. PA-VDM can be easily implemented on top of existing video diffusion models without changing their architectures. This allows for efficient training and generation of long videos, overcoming the limitations of previous approaches that require iterative generation of short clips.

**Weaknesses:**

An important baseline is missed.

The increasing-noise diffusion scheduler of PA-VDM is actually a multi-task training process, i.e., the model is trained on mixed types of conditions, and therefore the model can perform both video generation and video extending. Towards this goal, there is a more straightforward fine-tuning strategy than the increasing-noise diffusion scheduler, i.e., directly training the model on both text-to-video and video-to-video data. For example, if the maximum video length during training is L, and we can assign 10% training paris as generating L frames with only text without known frames, 10% pairs with text and one given frame, 10% pairs with text and two given frame, and so on.

PA-VDM needs to be compared with this baseline to show the priority of the increasing-noise diffusion scheduler.

**Questions:**

See the weakness above.

---

### Note · Authors · 2024-11-15

I have read and agree with the venue's withdrawal policy on behalf of myself and my co-authors.